# Mental Health Challenges of Young Labor Migrants from the Healthcare Professionals Perspective: Lessons Learned from a Multi-Country Meeting

**Daisy Michelle Princeton [1], Ida Marie Bregård [1], Marianne Annion [2], Gine Shooghi [3], Gitte Rom [3], Brynja Örlygsdóttir [4], Hildur Sigurðardóttir [4], Riita Kuismin [5], Joonas Korhonen [6] and Sezer Kisa [1,\*]**

1 Department of Nursing and Health Promotion, Oslo Metropolitan University, 0130 Oslo, Norway; dapri@oslomet.no (D.M.P.); idabre@oslomet.no (I.M.B.)
2 Department of Nursing, School of Health Care, Tallinn Health Care College, 13418 Tallinn, Estonia; marianne.annion@ttk.ee
3 Department of Nursing, University College Copenhagen, 1799 Copenhagen, Denmark; angb@kp.dk (G.S.); giro@kp.dk (G.R.)
4 Faculty of Nursing, University of Iceland, 101 Reykjavík, Iceland; brynjaor@hi.is (B.Ö.); hildusig@hi.is (H.S.)
5 Department of Rehabilitation and Health Care, South-Eastern Finland University of Applied Sciences-Xamk, FI-50101 Mikkeli, Finland; riitta.kuismin@xamk.fi
6 Faculty of Health and Well-Being, Turku University of Applied Sciences, 20520 Turku, Finland; joonas.korhonen@turkuamk.fi
\* Correspondence: sezkis@oslomet.no

**Abstract:** The mental health of young labor immigrants (YLI's) is a public health issue that has become notably more apparent during the COVID-19 pandemic. It is well established in the literature that most YLI's are young and healthy when they arrive in the host country. However, due to the poor living and working conditions, as well as linguistic and socioeconomic barriers to health care in the host country, their physical and mental health often deteriorates. Between 1 March 2021 and 5 March 2021, a virtual meeting was organized by Oslo Metropolitan University in collaboration with the Nordic Council of Ministers mobility and network program for education in the Nordic and Baltic countries (Nordplus). It consisted of a multidisciplinary team of 26 participants from Nordic and Baltic countries. Topics included working and living conditions of YLI's, prejudices towards immigrants, and mental health-related interventions for YLI's in the participating countries. This paper draws attention to some of the mental health challenges and needs of YLI's and to the suggestions gathered from the Nordplus meeting to combat these challenges from a healthcare professional's perspective.

**Keywords:** immigrants; health care students; mental health; young laborers

## 1. Introduction

Every day, people leave low-income countries to seek better jobs in more developed nations. Approximately 164 million migrant workers make up 4.7% of the global labor force [1]. Of these, more than half reside in North America, Western Europe, Eastern Europe, and high-income countries of the other regions [2]. In 2019, migrant workers living in the World Health Organization (WHO) European Region made up 17.8% of the working population, including 18% in Norway and 13.7% in Sweden [3–5].

Statistics show that young people migrate in larger numbers as they search for better-paid jobs [2,6,7]. This inflow of young workers is seen as a benefit by those countries with ageing populations because it expands the workforce and lowers the host country's dependency rate [2]. However, compared to native workers, labor immigrants are at higher risk of having insecure jobs with short-term contracts, low wages, and long working shifts with hard physical labor. They also often lack trade union representation regarding



occupational health and safety [2,6–10]. Even worse, economic turmoil during a crisis such as COVID-19 increases competition, inequality, and discrimination in the workplace due to limited job opportunities. All these factors may aggravate the mental health problems among migrant workers [10–12]. A recent study reported that YLI's are psychologically vulnerable [13], and most labor migrants fear being abandoned by their employers and stranded in a foreign country with no means of support [12].

## 2. Mental Health of Young Labor Immigrants

The literature on immigrants' mental health is extensive, indicating a strong association between being an immigrant and adverse mental health outcomes [3,5,14–16]. However, a study from China found that older labor migrants showed better mental health than younger migrants, suggesting the importance of mental health support of labor migrants [6].

Separation from family and social networks, lack of social support, difficulties in adapting to a new society, compounded by loneliness, discriminatory behaviour by the host society, and stigmatized identity [6,11,15,17–20], delays in obtaining legal employment status, uncertain socioeconomic status, legal uncertainties, and lack of knowledge about social rights all increase the risk of poor psychological wellbeing among YLI's [15–17].

Common emotional problems among YLI's include psychological distress, depression, anxiety, posttraumatic disorders, low self-esteem, emotional detachment, insomnia, increased use of alcohol and substances, decreased productivity, lack of motivation at work, and self-harm [8,14,19,21]. However, it is also reported that, unlike ethnic citizens, labor migrants receive little benefit from mental health services due to the fear of stigmatization related to mental problems, fear of losing the job, language barriers, the cost of health services, and healthcare workers' attitudes towards immigrants [4,8,22]. Indeed, one challenge identified by the Nordplus meeting participants was that health professionals have prejudices towards labor migrants and their mental health issues, which explains part of the hesitance to use mental health services.

## 3. Nordplus Meeting on the Mental Health of Young Labor Immigrants

A five-day online meeting on the mental health of YLI's, funded by Nordplus and organized by the Oslo Metropolitan University, was held between 1 March 2021 and 5 March 2021. Established by the Scandinavian Council of Ministers in 1988, Nordplus is a program that encourages cooperation within the field of education between the Nordic and Baltic countries by providing funding for networking, project collaborations, and exchanges. This program includes Denmark, Finland, Iceland, Norway, Sweden, the Faroe Islands, Greenland, and Åland. Latvia, Lithuania, and Estonia joined the network in 2008 [23]. The meeting aimed to create awareness among health care students about the mental health challenges of YLI's and share good practices regarding interventions to prevent mental health problems among labor migrants across the Nordic and the Baltic countries. Students and teachers of nursing and allied health care who were interested in exploring the mental health challenges of YLI's in the associated countries were invited to participate. In all, 15 students of nursing, social work, and paramedics and 11 nursing faculty members in the fields of mental health, psychiatry, medical and surgical, midwifery, and sociology participated in the meeting. Participants were from Denmark, Estonia, Finland, Iceland, Latvia, and Norway. The participating faculty members had advertised the online meeting in their schools and invited volunteers to participate in the meeting. Relevant information was given to the interested students, and verbal consent was obtained from all participating students. The objectives of the meeting were to (1) identify YLI mental health issues that can be prevented; (2) determine the main inhibiting and facilitating factors for YLI's to adjust to new conditions in the host country, and (3) share good practices of social/health care policy for the mental health of YLI's. During the meeting, the results of the literature review about the situation of young labor migrants' mental health across participating countries, social determinants of mental health, working and living conditions, prejudices towards

immigrants among health care professionals, the primary needs and challenges of YLI's, and the roles of health care professionals to combat mental health problems among YLI's were presented, followed by scholarly debates, panel discussions, and group works with specifically designed case studies. The anonymous online survey results for evaluation of the course showed that the course met the expectations of the participating students. The majority of the students strongly agreed (87%) that the course content enhanced their understanding of promoting the mental health of young labor immigrants.

## 4. Challenges and Suggestions for Health Care Professionals

The COVID-19 pandemic has raised concern about the mental health problems of immigrants, particularly labor immigrants, across the world [24]. The suggestions from this meeting and available studies demonstrate that during a crisis like COVID-19, health professionals are at the front line to serve vulnerable groups such as immigrants. However, their negative attitudes and behaviours towards immigrants are obstacles to utilizing mental health services [9,25–27]. This dilemma raises the question of how to fulfil the role expected of health professionals. The participants agreed that there is an urgent need for well-trained mental health professionals who are culturally sensitive and aware of age-specific mental health needs. The consensus was to train health professionals about culturally sensitive mental health care services throughout their undergraduate education. This suggestion was also supported by Illingworth. He emphasizes that the role of mental health nurses should be the construction of global mental health policies. He draws attention to the need to focus on sensitive, localized person-centred mental health care [24].

Young immigrants should receive mental health services from young health professionals. This suggestion is supported by the relevant literature [28,29]. Studies have shown that younger health professionals, particularly social workers, have fewer negative attitudes and prejudices towards immigrants [30,31].

The meeting participants also emphasized that an essential part of health care services should be the use of screening tools in primary care settings, in the field or through a doctor's referral for the early identification of mental health problems among labor immigrants. However, maintaining culture-friendly mental health services for labor immigrants demands highly qualified, culturally competent mental health care professionals [24]. According to Kirmayer et al., specific challenges in the mental health of immigrants include communication difficulties due to language and cultural differences [32]. Both Nordic and Baltic countries complained about problems in finding interpreters when caring for immigrants in the hospitals. Therefore, it was suggested that new technologies, including developing apps to screen for mental health problems of labor immigrants in different languages, are crucial.

## 5. Conclusions

In summary, this Nordplus meeting resulted in a cross-country understanding from the healthcare professionals' perspective of the mental health challenges of young labor migrants. These challenges reflected those mental health issues confronting labor immigrants around the world. It has emerged that there is a need for stronger cooperation in the training of healthcare professionals working in Nordic and Baltic countries on mental health.

Understanding the individual, social, workplace, and country-level roots of mental health challenges among YLI's is essential. Although previous studies revealed strong associations between mental health and other immigrant populations such as refugees and asylum seekers, this article highlights the lack of robust data on the mental health problems of young labor migrants in Nordic and Baltic countries.

It is well established in the literature that suspicion and hostility against immigrant workers are all too common. In contrast to Nordic countries, Baltic countries had additional challenges such as health care professionals' negative attitudes towards immigrants. These challenges need policy-level changes to protect the mental health of YLI's and their rights. Therefore, it was suggested at the meeting to establish support groups to strengthen social

networks and integration by coordinating activities with non-governmental organizations or voluntary institutions. Health policymakers are responsible for carrying out country-specific assessments to reveal barriers to low utilization of mental health services among labor migrants and taking necessary actions based on the assessment results.

This five-day meeting attempted to draw the attention of policymakers to distinguish immigrant groups (migrants, refugees, asylum seekers, undocumented immigrants, and labor immigrants) when planning interventions to prevent mental health problems. Further research is warranted regarding the prejudices of health care professionals against labor immigrants, specifically the young immigrants and their mental health challenges and health system response in support of young labor migrants in the Nordic and Baltic countries. It is highly recommended to organize such multi-country meetings for health care professionals and social policymakers to discuss the mental health challenges of labor migrants in other countries and regions worldwide.

**Author Contributions:** Conceptualization, D.M.P., I.M.B., M.A., G.S., G.R., B.Ö., H.S., R.K., J.K. and S.K.; methodology, D.M.P., I.M.B., M.A., G.S., G.R., B.Ö., H.S., R.K., J.K. and S.K.; resources, D.M.P., I.M.B., M.A., G.S., G.R., B.Ö., H.S., R.K., J.K. and S.K.; writing—original draft preparation, D.M.P., M.A., G.S., B.Ö. and S.K.; writing—review and editing, D.M.P. and S.K.; funding acquisition, D.M.P., I.M.B., M.A., G.S., G.R., B.Ö., H.S., R.K., J.K. and S.K. All authors have read and agreed to the published version of the manuscript.

**Funding:** This meeting was funded by NordPlus Higher Education (NPEH-2018/10058).

**Institutional Review Board Statement:** Not applicable.

**Informed Consent Statement:** Not applicable.

**Data Availability Statement:** Not applicable.

**Conflicts of Interest:** The authors declare no conflict of interest.

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
