# Peer review of "Mental Health Challenges of Young Labor Migrants from the Healthcare Professionals Perspective: Lessons Learned from a Multi-Country Meeting"

_sustainability, doi:10.3390/su131810482_

Round 1

Reviewer 1 Report

Dear authors

Greetings

I send you the attached doc to give you some suggestions. Please read it with Adobe. Regards

Author Response

I would like to thank the reviewer for his/her kind comments. The main concern of the manuscript is the mental health challenges of young labour immigrants. The manuscript was improved with the necessary changes, corrections and additional references and was colored in red throughout the manuscript. The manuscript was professionally proofread. 

Kind regards

Reviewer 2 Report

1. It would be helpful to state what the objective of this report is. It seems to summarize the discussion from the meeting, but I would urge caution with presenting it as research data as it does not appear ethical approval or informed consent was obtained. The summary of the findings is also very brief and I am not clear what this reports adds to the literature. 

2. Section 2 - the heading for this does not really match the rest of the paper. Was this a course for students or a meeting with stakeholders?

3. Much more information about the meeting is needed including an agenda, how data was analyzed, who attended the meeting, how was consensus reached, etc.

4. Was ethics approval obtained to use the data collected at the meeting?

Author Response

1-I would like to thank to the reviewer for his/her kind comments. The manuscript was improved as suggested, This commentary article is not research-based. Therefore, the necessary corrections were made regarding the methods of the manuscript.

2- This was a meeting with stakeholders. The necessary changes were made to the heading and the content.

3- Additional information was added about the meeting.

4-Verbal permission was obtained to disseminate the consensus reached in the meeting.

KInd regards,

Reviewer 3 Report

Dear Authors,

Thank you very much for providing a manuscript on this important topic. The manuscript would benefit from the revision.

Abstract: “26 participants of the meeting” – what are the characteristics of participants, are they scientist, from which field, or is it a multidisciplinary team? To add 1-2 sentence of participant characteristics would help readers understand the aim of the event. The aim is not clear: why creating awareness among health care students is important? Was the meeting mainly focused on how host countries can help labor migrants in terms of mental health prevention-treatment? It is suggested to clarify the aim of the event and the aim of this article.

Lines 29-33: This sentence is very long and contains multiple aspects. The authors may consider splitting it into several sentences and giving explanation what they mean in ‘lack of access to trade union rights’

Lines 41-42: ‘a strong association between being an immigrant and adverse mental health outcomes’- do you mean all kinds of migrants or a specific group (labor, refugees etc)?

Lines 54-55: „a Chinese study“- suggested to change: A study from China  

Lines 61-62:”However, it is also clear that labor migrants do not particularly benefit from mental health services than ethnic citizens for well-known reasons” – What do you mean in ‘well-known reasons’?

Line 70: After lit. review on mental health impacts of international migrants, to move directly to ‘Intensive Course for Health Care Students’ as a new chapter would not be clear for the readers. The authors may consider to give more clear sub-title: e.g. lessons learned from Nordic Countries, or good example of… interventions for migrant mental health: example from NordPlus

Line 71: The authors are suggested to give more information on NordPlus. Is it a governmental organization, what is a main focus of this organization?

Line 126: The authors may consider clarifying the meaning of ‘narrow research results’

Line 127: The authors may consider clarifying the meaning of ‘contradicting factors’

In addition to the already mentioned aspects, authors may consider discussing more about health system response in support of young labor migrant from the perspective of host countries.

Author Response

We appreciate the reviewer`s recommendation.

1-           Additional information was added about the characteristics of participants. The aim of the article was clarified, and necessary changes were made throughout the manuscript.

2-           Lines 29-33: The sentence was changed as suggested, and a clarification sentence was added for ‘lack of access to trade union rights. The changes were made in red.

3-           Lines 41-42: Necessary changes were made for clarification.

4-           Lines 54-55: „a Chinese study“-changed to A study from China.

5-             The necessary information was added to clarify ‘well-known reasons’?

6-           Thank you for the constructive comments. The title has been changed

7-           Additional information was added about NordPlus.

8-           Necessary corrections were made.

9-           Necessary changes were made to clarify the meaning of ‘contradicting factors’

10-        We appreciate the reviewer`s comment. Future research is suggested to explore the health system response in support of young labour migrants in the Nordic and Baltic countries.

Best regards, 

Round 2

Reviewer 1 Report

Dear Authors  Greetings

I appreciate your efforts. The conclusions would can make a point about covid19 effects and perspectives of risks, future advances and the importance to apply the same model in others countries and regions worldwide. Congrats. Regards

Author Response

We appreciate the reviewers recommendation. Suggested points were added in the conclusion section and the changes were highlighted in red.

Best regards, 

Reviewer 2 Report

the authors have addressed my concerns. 

Author Response

We appreciate his/her comments. 

Best regards, 

Reviewer 3 Report

Dear Authors,

Thank you very much for providing updated version of the manuscript. The manuscript has improved considerably. I would suggest one minor edit only. See below:

Abstract:  For 'Nordplus' – please give full name first and then, use acronym (similar to YLI that you already did). Readers may not be familiar what Nordplus means.

Author Response

I would like to thank to the reviewer for his/her kind comments. The clarification was done about "Nordplus".

Best regards,